# Antibacterial Evaluation of Zirconia Coated with Plasma-Based Graphene Oxide with Photothermal Properties

**DOI:** 10.3390/ijms24108888

**Published:** 2023-05-17

**Authors:** Lydia Park, Hee-Seon Kim, Woohyung Jang, Min-Kyung Ji, Je-Hwang Ryu, Hoonsung Cho, Hyun-Pil Lim

**Affiliations:** 1Department of Prosthodontics, School of Dentistry, Chonnam National University, 33 Yongbong-ro, Buk-gu, Gwangju 61186, Republic of Korea; piijoy12@hanmail.net (L.P.);; 2Dental 4D Research Center, Chonnam National University, 33 Yongbong-ro, Buk-gu, Gwangju 61186, Republic of Korea; 3Department of Pharmacology and Dental Therapeutics, School of Dentistry, Chonnam National University, 77 Yongbong-ro, Buk-gu, Gwangju 61186, Republic of Korea; 4School of Materials Science & Engineering, Chonnam National University, 77 Yongbong-ro, Buk-gu, Gwangju 61186, Republic of Korea

**Keywords:** graphene oxide, atmospheric plasma, antibacterial surface, photothermal antibacterial effect

## Abstract

The alternative antibacterial treatment photothermal therapy (PTT) significantly affects oral microbiota inactivation. In this work, graphene with photothermal properties was coated on a zirconia surface using atmospheric pressure plasma, and then the antibacterial properties against oral bacteria were evaluated. For the graphene oxide coating on the zirconia specimens, an atmospheric pressure plasma generator (PGS-300, Expantech, Suwon, Republic of Korea) was used, and an Ar/CH_4_ gas mixture was coated on a zirconia specimen at a power of 240 W and a rate of 10 L/min. In the physiological property test, the surface properties were evaluated by measuring the surface shape of the zirconia specimen coated with graphene oxide, as well as the chemical composition and contact angle of the surface. In the biological experiment, the degree of adhesion of *Streptococcus mutans* (*S. mutans*) and *Porphyromonas gingivalis* (*P. gingivalis*) was determined by crystal violet assay and live/dead staining. All statistical analyzes were performed using SPSS 21.0 (SPSS Inc., Chicago, IL, USA). The group in which the zirconia specimen coated with graphene oxide was irradiated with near-infrared rays demonstrated a significant reduction in the adhesion of *S. mutans* and *P. gingivalis* compared with the group not irradiated. The oral microbiota inactivation was reduced by the photothermal effect on the zirconia coated with graphene oxide, exhibiting photothermal properties.

## 1. Introduction

Peri-implantitis is the most common cause of early or late implant failure [1]. The main cause of peri-implantitis has been found to be microbial colonization [2,3,4], in which bacteria irreversibly agglomerate and attach to the surface of a tooth or implant to form a bacterial biofilm. If these biofilms continue to accumulate, dental plaque is formed, and when this plaque accumulates on the implant surface, it can cause peri-implantitis and loss of the alveolar bone [5,6,7].

In clinical practice, several antibacterial therapies are used to kill accumulated bacteria. In general, mechanical debridement by scaling and root planning (SRP) to remove the subgingival plaque is commonly performed in nonsurgical periodontal therapy. However, this cannot completely remove the subgingival plaque [8,9]; thus, several antibiotics are used to remove residual bacteria [10]. Unfortunately, the overuse of antibiotics results in the emergence of drug-resistant bacteria, rendering antibiotic treatment ineffective. As such, it is important to develop a new strategy to not only eliminate residual bacteria in the early stages of treatment, but also to maintain a low bacterial count in the long term.

Antibacterial photothermal therapy is a method that effectively kills bacteria through various thermal effects resulting from the heat that photothermal agents generate under near-infrared irradiation [11]. Near-infrared irradiation used in photothermal therapy can penetrate deep into the tissue and effectively kill bacteria with little photodamage [12,13]. Antibacterial photothermal therapy requires photothermal agents, which are mediators that generate heat by absorbing energy from near-infrared irradiation. The heat generated in photothermal therapy is caused by electromagnetic waves; when photothermal agents are irradiated with near-infrared light, they generate heat by absorbing the remaining incident photons after some are scattered. The absorbed photons are involved in heat generation and light emission, and a photothermal effect occurs from the generated heat [14]. 

Typical photothermal agents for antibacterial photothermal therapy include gold nanoparticles, conductive molecules, and graphene-based materials [15,16,17,18]. Graphene has a single-layer honeycomb lattice structure of aromatic carbon atoms, and unique physicochemical properties [19,20,21]. Graphene oxide (GO), which contains abundant functional groups such as carboxyl, hydroxyl, and epoxy groups, is characterized by high mechanical flexibility and hydrophilicity. Moreover, because of its close electron energy levels, GO boasts remarkable light absorption, resulting in high light-to-heat conversion efficiency and excellent biocompatibility [22,23]. These properties have resulted in a rising number of studies using GO as an effective photothermal agent in photothermal therapy. Omid et al. [24] used reduced graphene oxide nanomesh (rGONM), as one of the recent structures of graphene with a surprisingly strong near-infrared (NIR) absorption, is used for achieving ultraefficient photothermal therapy. The excellent NIR absorbance and tumor targeting of rGONM-PEG-Cy7-RGD results in an ultraefficient photothermal therapy (100% tumor elimination 48 h after intravenous injection of an ultralow concentration (10 μg mL^−1^) of rGONM-PEG-Cy7-RGD followed by irradiation with an ultralow laser power (0.1 W cm^−2^) for 7 min).

Reduced graphene oxide (rGO) is a promising alternative for the bulk production of graphene-like materials. The bottleneck of its commercialization is the control of oxygen functional groups on the surface to engineer its diverse properties, such as electronic structure, optical properties, and surface properties [25]. Both graphene oxide (GO) and reduced graphene oxide (rGO) effectively absorb near-infrared (NIR) light, which is a biocompatible light source that penetrates tissues. Moreover, GO and rGO convert the absorbed NIR light energy to heat, increasing the temperature in GO and rGO and their surrounding media [26,27]. While both GO and rGO can absorb NIR, rGO is more effective [28], likely because of the red shift in the absorbance peak from approximately 230 to 260 nm [26,29]. In a related study, the rGO synthesized in this study showed about 10 times higher absorbance than GO at 880 nm. The reason for the small temperature rise in GO was that its absorbance at 800 nm was significantly lower than that of other samples [30].

Conventional methods of manufacturing and coating with rGO have disadvantages that include susceptibility to contamination, the potential of contamination by residue from the solution used during manufacturing, and the generation of harmful gas [31,32,33]. In this study, rGO was coated using plasma to overcome these disadvantages. Plasma is a gas that is ionized and charged with energy, and treating living tissues with plasma can change their wettability, as well as their mechanical and biological properties [34]. The proposed method of coating graphene with plasma has the advantage of being simple and reasonable in terms of cost, while not requiring other additives or generating byproducts during the production of rGO [35].

Because zirconia implants have excellent biocompatibility and aesthetics, the range of their application has been gradually expanding [36,37]. Furthermore, zirconia implants not only have the ability to inhibit the level of bacterial adhesion but can also be applied to patients with a titanium allergy [38]. Coating the surface of zirconia with rGO using plasma can enhance the biocompatibility and antibacterial properties of the zirconia implant. in addition, when peri-implantitis occurs around zirconia implants, the possibility of treatment can be improved through antibacterial photothermal therapy by irradiating with near-infrared radiation. Yi et al. (2020) reported that antibacterial photothermal therapy enabled the efficient killing of bacteria through various heat effects, which resulted from the heat generated by photothermal agents under near-infrared irradiation [11].

Therefore, in this study, we coated zirconia specimens with rGO with photothermal properties using atmospheric plasma, and we confirmed the effect of reducing bacterial biofilms in *S. mutans*, which are involved in the formation of the initial biofilm on the surface of zirconia implants and *P. gingivalis*, a bacterium that causes peri-implantitis.

## 2. Results

### 2.1. Surface Characteristics

The surface aspects of the zirconia specimens and reduced graphene oxide (rGO)-coated specimens were observed through a field-emission scanning electron microscope (FE-SEM). For the ZG group, a cloud-shaped substance was observed to spread over the surface, which is presumed to be rGO (Figure 1b). 

Raman spectroscopy revealed the typical peaks for graphene are D (1350 cm^−1^), G (1580 cm^−1^), and a 2D peak at 2690 cm^−1^. Pristine graphene does not show any D peak, which represents edges of a graphene crystal and chemical bonds [39], while typical Raman spectra for GO are characterized by its D and G band corresponding to 1353 cm^−1^ and 1605 cm^−1^, respectively [40]. The famous bands of graphene sheets in Raman spectra, i.e., the G band (~1582 cm^−1^) originated from the first-order scattering of the E2g phonons of sp2-hybridized carbon atoms, the D band (~1350 cm^−1^) caused by a breathing mode of j-point phonons of A1g symmetry of the defects involved in the sp3 -hybridized carbon bonds such as hydroxyl and/or epoxide bonds [41], and the 2D band (~2679 cm^−1^) which is much sensitive to stacking of graphene sheets [42] are observable in the spectra shown in Figure 1c. Compared with the Zr group, the Zr-rGO group showed an increase in the contact angle (Figure 1d,e). 

### 2.2. Photothermal Effect under 940 nm Laser Irradiation

As shown in Figure 2, the surface temperature of the graphene-oxide-coated zirconia specimens rapidly increases during NIR irradiation with different power intensities, and stabilizes after 120 s of irradiation. On the other hand, the surface temperature of the zirconia specimens without graphene oxide coatings was observed to rise slightly. Additionally, it was found that the higher the laser output power, the steeper the temperature rise. 

### 2.3. Inhibition of Biofilm Formation 

#### 2.3.1. Temperature Rise

The specimens were irradiated at 4 W and 120 s (480 J) using a near-infrared laser. Of the four groups (Z, ZG, ZN, and ZGN), the ZN group, which was irradiated with NIR on the zirconia surface, and the ZGN group, which was irradiated with NIR rays on the reduced-graphene-oxide-coated zirconia surface, were observed to show an increase in surface temperature (Figure 3a). The average temperature of the ZN group increased from 25.8 °C to 36.8 °C, a rise of 11 °C, while that of the ZGN group increased by 30.9 °C, from 25.8 °C to 56.7 °C, which is a greater temperature rise than that of the ZN group (Figure 3b).

#### 2.3.2. Evaluation of Adhesion Ability of Oral Bacteria after NIR Irradiation

The adhesion of *S. mutans* for the ZGN group significantly decreased compared with the Z, ZG, and ZN groups (*p* < 0.008). For the ZN group, the adhesion of *S. mutans* was observed to increase compared with the ZGN group (Figure 4a). When the LIVE/DEAD^®^ BacLight™ Bacterial Viability Kit was used, live bacteria were observed as green. It was observed that the rate of *S. mutans* and *P. gingivalis* death was high in the ZGN group (Figure 5).

The adhesion of *P. gingivalis* significantly decreased in the ZGN group compared with the Z group (*p* < 0.001). Even compared with the ZG and ZN groups, the adhesion of *P. gingivalis* significantly decreased in the ZGN group (*p* < 0.05) (Figure 4b). 

## 3. Discussion

The average prevalence of peri-mucositis and peri-implantitis in the dental field was reported to be 46.83% and 19.83%, respectively [43]. The main etiology is due to bacterial biofilms formed on the surface of implants [44]. To remove these biofilms, mechanical debridement is used as a standard treatment [45]. However, this treatment can be beneficial for the treatment of peri-mucositis, while its effectiveness in the treatment of peri-implantitis may be limited [46]. To date, the treatment of peri-implantitis remains a challenge in the dental field, with no consensus on an appropriate strategy [47]. For the treatment of peri-implantitis, various treatments that focus on bacterial reduction, including oral hygiene education and antibacterial therapy, have been recommended [48]. 

In addition, research has shown that photothermal therapy in dental applications reduces bacterial growth. Antibacterial photothermal therapy (aPTT) has emerged as a potential alternative to antibiotics to reduce oral microbes [49]. This is a non-invasive treatment method with repeatable target specificity, in which bacterial resistance is unlikely to develop [50,51].

There are well-known mechanisms involved in the antibacterial activity of graphene are the physical direct interaction of extremely sharp edges of nanomaterials with cell wall membrane [52], generating reactive oxygen species (ROS) under visible light in air [53], generating ROS in absence of light [54], trapping the bacteria within the aggregated nanomaterials [55], oxidative stress [56], interruption in the glycolysis process of the cells [57], DNA damaging [58], heavy metal ion release [59], and recently contribution in generation/explosion of nanobubbles) [60]. And graphene also showed excellent photothermal antibacterial performance in response to NIR light, which is a biocompatible light source that penetrates tissues. The NIR laser photothermal therapy can focus on a targeted area for effective treatment. Photothermal bactericidal surfaces based on hyperthermia provide a biocide-free and high-efficiency way to kill surface-attached bacteria, in particular, the multidrug-resistant bacteria. The bacteria-killing mechanism of this thermal effect was resulted from both the hyperthermia-induced protein denaturation and subsequent bacterial thermal decomposition [61], and disrupting the bacterial membrane to facilitate the permeation of ROS into bacteria [11]. In previous study, A graphene-based photothermal agent was efficiently capturing and killing of both gram-positive *Staphylococcus aureus* (*S. aureus*) and gram-negative *Escherichia coli* (*E. coli*) bacteria upon near-infrared (NIR) laser irradiation [62].

Although the remarkable achievements of photothermal bactericidal surfaces have been made, it should be noted that there are still some challenges related to this promising field and will be left. One of that is taking the standards of simple, economic, and environmental-friendly manufacturing methods into consideration for the surface design to accelerate the practical process. In this study, graphene was coated using plasma to overcome these disadvantages. Plasma is a gas that is ionized and charged with energy and treatment of living tissues with plasma can change the wettability, as well as their mechanical and biological properties [34]. The proposed method of coating graphene with plasma has the advantage of being simple and reasonable in terms of cost, while not requiring other additives or generating by-products during the production of GO [35].

With this surface-coating method, reduced graphene oxide without deformation was deposited on the surface simply and quickly. In the results, the adhesion of *S. mutans* and *P. gingivalis* was not significantly reduced when comparing the Z and ZG groups. However, when the ZG group and the ZGN group were compared, a significant reduction was obtained.

In this paper, reduced graphene oxide (rGO) was used as a light-activated heating nanoparticle (L-HNP) for its photothermal effect in addition to its antibacterial properties. In the results, the adhesion of *S. mutans* and *P. gingivalis* was not significantly reduced when comparing the Z and ZG groups. However, when the ZG group and the ZGN group were compared, a significant reduction was obtained. Bacteria in the mouth begin to die at temperatures above 60 °C [63]. However, in this paper, the antimicrobial activity by the shape of graphene oxide itself was used together with a temperature of 55–60 °C to confirm the activity of bacteria. In the results of this experiment, it can be considered that the reason why the bacterial activity only fell to about 50% was because the experiment was not conducted at a high temperature of 60 °C or more. 

It is expected that the bacterial activity will be much lower when tested at a temperature of 60 °C or higher. However, temperatures above 60 °C can effectively reduce bacteria, but can cause thermal damage to normal cells. Therefore, research that can reduce bacteria without affecting normal cells is needed. In order to reduce this thermal damage, related studies was to compare the effect of treatments with carcumin-mediated aPDT and chlorhexidine diguluconate (CHX) in relation to the viability of specific microorganisms. As a result, the combination of photodynamic therapy and chlorhexidine resulted in more bacterial reduction [64]. Therefore, research is needed to confirm the combined effect of several drugs at an appropriate temperature that can cause thermal damage to bacteria while reducing thermal damage to normal cells.

As a result of the surface angle, it was observed that the graphene oxide was coated with hydrophobicity. In a related paper studying the hydrophobic properties of graphene oxide, the adhesion of bacterial cells was prevented, and furthermore, the hydrophobic interaction was found to destroy the bacterial membrane, resulting in antibacterial action [65]. In this study, the Zr group demonstrated a relatively high hydrophilicity (low angle). This surface characteristic seemed to affect the early adhesion of cells, while coating zirconia with GO did not affect early adhesion and improved cell proliferation and differentiation.

In this experiment, when the zirconia specimens that were not coated with rGO were irradiated with NIR rays, it was observed that the surface temperature increased only up to 36–40 °C, leading to increased bacteria. On the other hand, for the rGO-coated specimens, the temperature increased to 56–60 °C under the same conditions, resulting in a greater reduction in bacterial adhesion due to higher temperatures. Based on this, it can be said that rGO is an excellent photothermal agent, as it rises to a high temperature upon NIR irradiation.

The NIR wavelength of 940 nm used in this experiment was reported to inhibit interleukin-6, monocyte chemotactic protein-1, interleukin-10, and tumor necrosis factor-α, thereby acting as an anti-inflammatory, while healing wounds, relieving pain, and reducing swelling [66,67]. Such anti-inflammation and wound healing effects of this NIR wavelength can produce positive results in the treatment of peri-implantitis along with antibacterial photothermal therapy through the increase in surface temperature. Because this study is an in vitro experiment, there are limitations in its clinical applicability. Zhuqing et al. suggested that irradiation conditions should be adjusted in in vivo experiments after in vitro experiments to treat patients using a photothermal agent through NIR irradiation [68]. If the bacterial reduction is confirmed by adjusting the irradiation conditions in in vivo experiments, it will be possible to reduce the mechanical damage around the implants caused by photothermal therapy and treat peri-implantitis through NIR irradiation.

## 4. Materials and Methods

### 4.1. Experimental Materials

#### 4.1.1. Specimens

Zirconia specimens (Zirmon^®^, Kuwotech, Gwangju, Republic of Korea) were prepared with dimensions of 15 mm diameter × 2.5 mm thickness. To render the surface of the specimens uniform, a grinder (LaboPol-5, Struers Co., Guiseley, UK) was used, and the surface was ground underwater with #800 SiC abrasive paper. All specimens were cleaned in an ultrasonic cleaner for 20 min each, using acetone, alcohol, and distilled water, in that order. After drying at room temperature, the specimens were sterilized in an autoclave (HS-3460SD, Hanshin Medical Co., Incheon, Republic of Korea).

#### 4.1.2. Reduced Graphene Oxide (rGO) Deposition

Reduced graphene oxide (rGO) was coated on the zirconia specimens using an atmospheric-pressure plasma generator (PGS-300, Expantech Co., Suwon, Republic of Korea). After mixing argon gas (4 L/min) and methane gas (3.5 L/min) in a quartz tube, the high-frequency (900 MHz) plasma generator was used to coat the specimens at a rate of 10 L/min and 240 W. The specimens were fixed with a circular clamp, and when the plasma was being applied, the distance between the plasma flame and the specimen was maintained at 25 mm. Additionally, the reciprocation from side to side of the clamp, which was simultaneously rotated, ensured that the application of rGO was uniform over the specimens. The plasma was applied for a total of 1 min per specimen by rotating it at 180 rpm and setting it to reciprocate 4 times for 15 s each time (Table 1).

### 4.2. Assessment of Surface Characteristics

The formation of micropores and nanopores on the surface structure of the samples with rGO were analyzed using a field emission scanning electron microscope (FE-SEM; S-4700, Hitachi, Japan). Raman spectroscopy of the sample was performed to determine the status of the graphene oxide deposition using a laser Raman spectrophotometer (NRS-5100, JASCO, Seoul, Republic of Korea) at a laser excitation of 532.13 nm. The wettability, a critical factor for biocompatibility, was determined via contact angle measurements using a video contact angle measuring device (Phoenix 300, SEO, Suwon, Republic of Korea).

### 4.3. Photothermal Effects under 940 nm Laser Irradiation

The photothermal conversion efficiency of graphene oxide was measured using the following method. The NIR laser light (940 nm) at a power of 2.5 W/cm^2^, 3.0 W/cm^2^, 3.5 W/cm^2^, and 4.0 W/cm^2^ was focused to a spot size of 1.5 cm. The graphene-oxide-coated specimen was swelled in phosphate-buffered saline (PBS) to reach equilibrium under light irradiation, and the temperature was recorded at 20 s intervals for a total of 5 min using a thermal imager (FLIR, E40, with an accuracy of 0.1 °C).

### 4.4. Assessment of Bacterial Activity

#### 4.4.1. Bacterial Culture

*Streptococcus mutans* (KCOM 1054, Gwangju, Republic of Korea), a Gram-positive bacterium known to be involved in the early stage of biofilm formation, and *Porphyromonas gingivalis* (KCOM 2804, Gwangju, Republic of Korea), a Gram-negative anaerobic bacterium known to cause peri-implantitis, were obtained from the Korean Collection for Oral Microbiology (KCOM). *S. mutans* strains were cultured at 37 °C in a culture chamber (LIB-150M, DAIHAN Labtech Co., Namyangju, Republic of Korea) using a brain–heart infusion (BHI; Becton, Dickinson and Company, Sparks, MD, USA) medium. *P. gingivalis* strains were also cultured at 37 °C in an anaerobic culture chamber (Forma Anaerobic System 1029; Thermo Fisher Scientific, Waltham, MA, USA) using a tryptic soy broth (TSB; Becton, Dickinson and Company, Sparks, MD, USA) medium.

#### 4.4.2. Bacterial Inoculation

All samples were sterilized in an autoclave (HS-3460SD, Hanshin Medical Co., Incheon, Republic of Korea) for 2 h. The samples for each group were prepared and fixed on a 24-well plate (SPL Life Sciences Co., Ltd., Pocheon, Republic of Korea). Each sample was inoculated with *S. mutans* and *P. gingivalis* (1.5 × 10^7^ CFU/mL) and cultured for 24 h and 48 h, respectively.

#### 4.4.3. In Vitro Antibacterial Effects

The in vitro antibacterial effects of photothermal therapy against *S. mutans* and *P. gingivalis* were evaluated. A 500 μL bacterial suspension was cultured for each condition. A total of 32 zirconia specimens were prepared for each bacterial sample; of these zirconia specimens, 16 were coated with rGO using atmospheric-pressure plasma. Depending on whether a rGO coating and/or near-infrared irradiation was applied, the specimens were divided into 4 groups (Z: zirconia, ZG: zirconia coated with graphene oxide, ZN: zirconia irradiated with NIR, and ZGN: zirconia coated with graphene oxide and irradiated NIR groups), each with 8 specimens.

All specimens were sterilized and then tested. First, all specimens were placed in a 24-well plate, in which 1 mL of *S. mutans* diluted to a bacterial concentration of 1.5 × 10^7^ CFU/mL was dispensed, and cultured for 24 h under aerobic conditions. After 24 h of incubation, the ZN and ZGN groups were fixed at a distance of 15 mm from the irradiation point to their surface and irradiated with near-infrared (NIR) at a power of 4 W for 120 s (480 J). The surface temperature generated in the process was measured with a thermal imaging camera (FLIR, E40, with an accuracy of 0.1 °C) at a distance of 20 cm from the surface. After irradiation, the bacteria on the specimens were further cultured for 24 h, and then the specimens, with the culture medium in the 24-well plate removed, were washed with PBS.

For the effect test against *P. gingivalis*, zirconia specimens were placed in a 24-well plate, in which 1 mL of *P. gingivalis* diluted to a bacterial concentration of 1.5 × 10^7^ CFU/mL was dispensed, and cultured for 72 h under anaerobic conditions. After incubation, the specimens were NIR irradiated at a power of 4 W and 120 s (480 J), and then the bacteria on the specimen were further cultured for 24 h. Afterward, bacterial adhesion evaluation was performed in the same manner using a crystal violet assay. The specimens were stained with a 0.3% crystal violet solution for 10 min, and washed twice with PBS after aspirating and removing the solution. Specimens were decolorized with 400 μL of a destaining solution, which consisted of 80% ethanol and 20% acetone, for 10 min, and 200 μL of the used solution was transferred to a 96-well plate using a pipette. Finally, the plate was read at 595 nm on a microplate reader (VersaMax™, Molecular Device LLC, San Jose, CA, USA). The degree of bacterial adhesion of *S. mutans* and *P. gingivalis* was visually evaluated using the LIVE/DEAD^®^ BacLightTM Bacterial Viability Kit (SYTO 9^®^, Molecular Probes Europe BV, Leiden, The Netherlands). Experiments were performed on *S. mutans* and *P. gingivalis*, and a total of 12 zirconia specimens were prepared for each bacterium. Six of these were coated with rGO using atmospheric-pressure plasma, and the twelve specimens were divided into four groups (Z, ZG, ZN, and ZGN) of three specimens each, depending on whether a rGO coating and/or NIR irradiation was applied. After culturing the bacteria, the culture medium in which the floating bacteria remained was washed with PBS solution. Each specimen in a well plate was injected with 200 μL of fluorescence reagent (SYTO 9 dye:propidium iodide:dH_2_O 1.5 μL:1.5 μL:1.0 mL), and stained at room temperature for 15 min with the plate covered with aluminum foil to prevent light from entering. Next, the remaining staining solution was washed with PBS solution, and the bacteria adhered to the specimens were observed using confocal laser scanning microscopy (Leica TCS SP5 AOBS/tandem, Leica Microsystems, Wetzlar, Germany).

### 4.5. Statistical Analysis

Statistical analysis of the crystal violet assay results was performed using the SPSS 21.0 software package (SPSS Inc., Chicago, IL, USA). The crystal violet assay results of *P. gingivalis* were assumed to have equal variances through Levene’s test and satisfied the normality requirement; therefore, they were statistically analyzed by a one-way ANOVA, which is a parametric method, and post-tested using the Tukey test. All results were tested for significance at the *p* < 0.05 level.

The crystal violet assay results of *S. mutans* were not assumed to have equal variances according to Levene’s test and did not satisfy the normality requirement; therefore, the results were statistically analyzed using the Kruskal–Wallis H test, which is a non-parametric method, and post hoc tested using the Bonferroni correction method. All results were tested for significance at the *p* < 0.008 level.

## 5. Conclusions

The results of this study indicate that when zirconia specimens coated with graphene oxide with photothermal properties are irradiated with NIR rays, the adhesion of *S. mutans* and *P. gingivalis* is reduced, suggesting that photothermal therapy can be effectively used in treatment to reduce oral bacteria. These findings can be used to prevent peri-implantitis and enhance therapeutic effects.

## Figures and Tables

**Figure 1 ijms-24-08888-f001:**
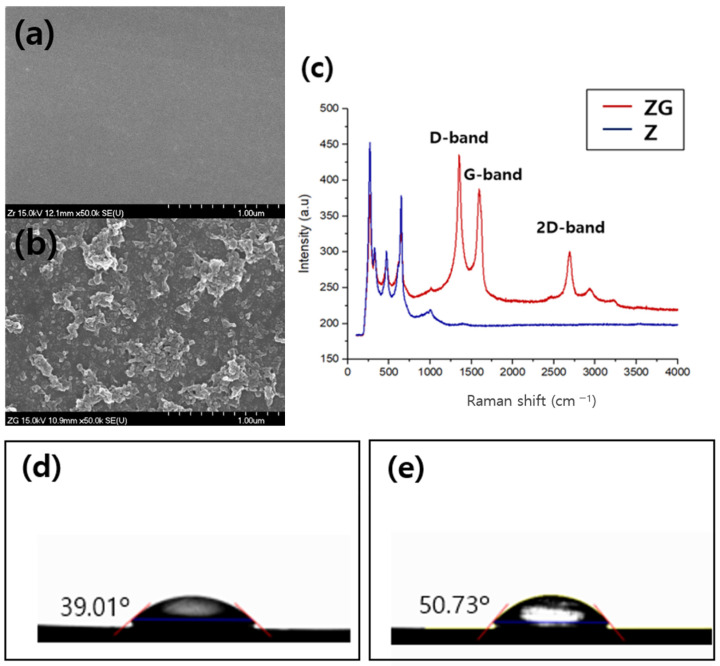
Characterization of reduced graphene oxide (rGO). (**a**) SEM image of surface morphology of Zirconia surface, (**b**) rGO-coated zirconia (10.9 mm, ×50,000). (**c**) Raman spectrum of rGO (red) and without graphene oxide (blue) (Z: Zirconia, ZG: Zirconia coated with rGO). (**d**) Water droplets on surface of the contact angle. Control (Zr) group (**e**) rGO-coated zirconia (Zr-rGO) group.

**Figure 2 ijms-24-08888-f002:**
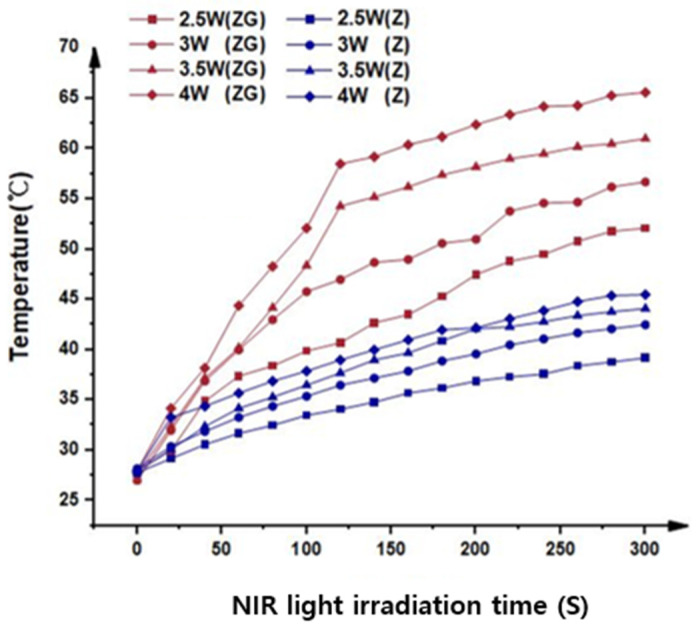
Photothermal properties under 940 nm laser irradiation at different powers. The surface temperature of the zirconia coated with rGO specimens rapidly increases during NIR irradiation (Z: Zirconia, ZG: Zirconia coated with rGO).

**Figure 3 ijms-24-08888-f003:**
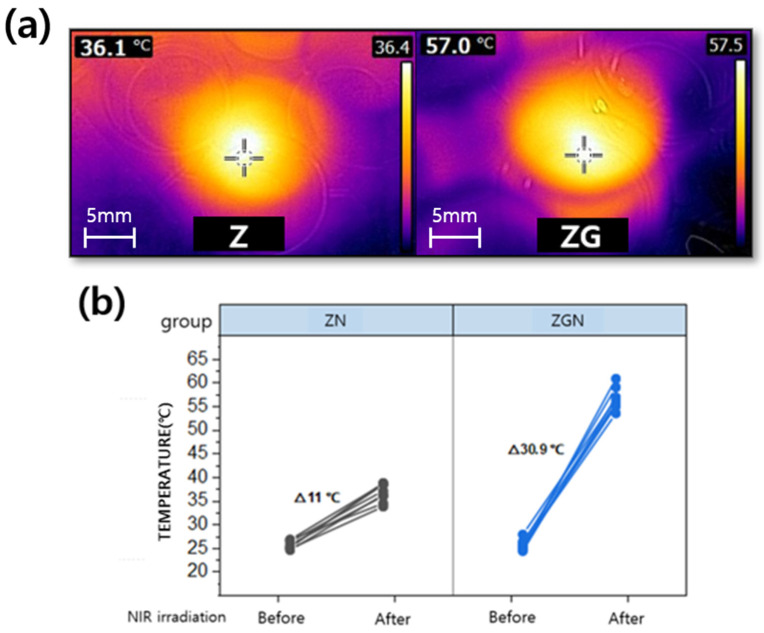
(**a**) Sample irradiated by NIR captured by a near infrared camera: Z (zirconia specimen irradiated with NIR), ZG (zirconia specimen coated with reduced graphene oxide irradiated with NIR); (**b**) Temperature change by NIR irradiation on zirconia (ZN) and zirconia coated with reduced graphene oxide (ZGN).

**Figure 4 ijms-24-08888-f004:**
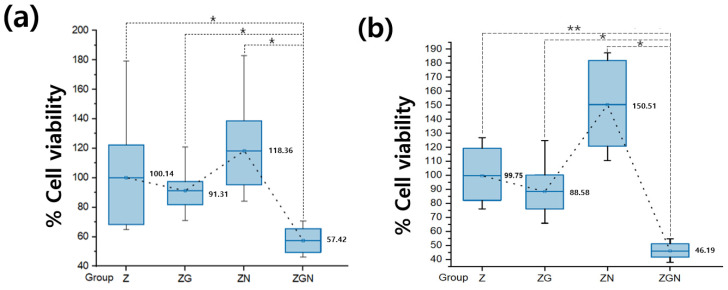
(**a**) The effect of photothermal treatment for the attachment of *Streptococcus mutans and* on zirconia specimens (control group), zirconia coated with reduced graphene oxide (ZG), zirconia irradiated with NIR (ZN group), and zirconia coated with reduced graphene oxide irradiated with NIR (ZGN group) (*n* = 8) (the result of Kruskal–Wallis test, *: marginally significant at *p* < 0.008). (**b**) The effect of photothermal treatment for the attachment of *Porphyromonas gingivalis* (*n* = 8) (the result of one-way ANOVA test, *: marginally significant at *p* < 0.05, **: marginally significant at *p* < 0.001).

**Figure 5 ijms-24-08888-f005:**
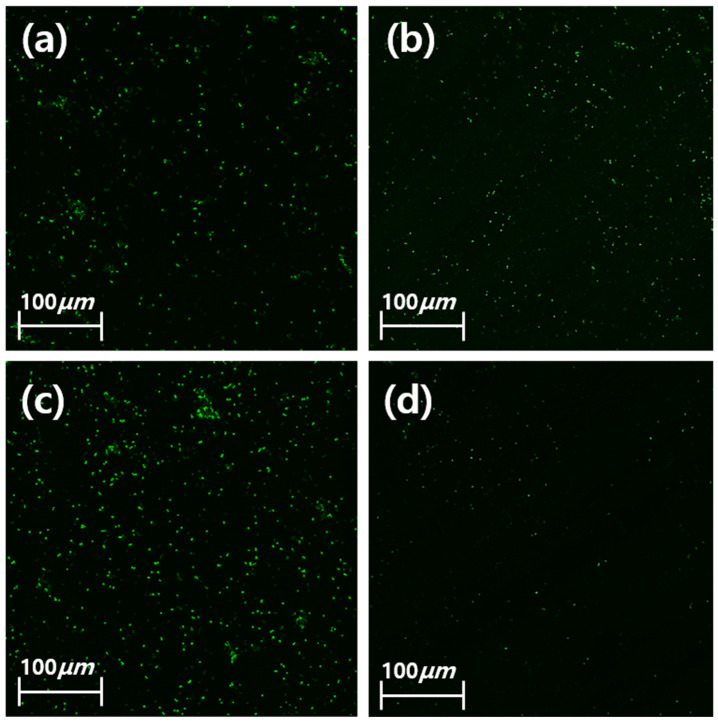
Viability of the *oral bacterium* on the (**a**) Z group, (**b**) ZG group, (**c**) ZN group, and (**d**) ZGN group (*n* = 3). Green fluorescence indicates viable cells.

**Table 1 ijms-24-08888-t001:** Parameters of atmospheric plasma generator.

Parameter	Value
Average working power (W)	300
Voltage (V)	27
Frequency (MHz)	900
Atmospheric pressure (Torr)	760
Electrode type	Electrodeless
Cooling type	Air cooled
Plasma density	10^15^/cm^3^

## Data Availability

Not applicable.

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
