# Peer review of "Antibacterial Evaluation of Zirconia Coated with Plasma-Based Graphene Oxide with Photothermal Properties"

_ijms, 2023, doi:10.3390/ijms24108888_

Round 1

Reviewer 1 Report

1.      The introduction section needs to be divided into paragraphs.

2.      Authors should provide some surface characterization of zirconia before and after graphene oxide coating. Such as appearance, SEM, XPS, contact angle et al.

3.      Why did the temperature of zirconia after NIR irradiation rise? What is the mechanism?

4.      Why zirconia with NIR irradiation can promote cell viability?

5.      If the coating amount of graphene oxide increase or decrease, does the temperature and cell viability change under NIR irradiation? Authors should try some various parameter in this study.

6.      How to calculate the cell viability?

Author Response

"Please see the attachment

Reviewer 2 Report

The submitted manuscript reports on the antibacterial evaluation of zirconia coated with plasma-based 2 graphene oxide with photothermal properties. The quality of the paper was poor and some of the fluorescence imaging data needs to be questioned, and therefore the manuscript cannot be published in the International Journal of Molecular Sciences.

1.      The introduction is suggested to be split into different paragraphs to make the logic clearer.

2.      Fig. 1D cannot be found in Figure 1 but it exists on line 107.

3.      The merged images should be provided in Figure 5 and Figure 7. Why is the same area dyed red and green? The reliability of the data needs to be questioned

4.      Where is the conclusion?

Reviewer 3 Report

Antibacterial performance of the graphene oxide/zirconia-based coating exposed to NIR irradiation has was studied. The subject is interesting and valuable. However, there are some important points which should be addressed, clarified and discussed in the revised version. Therefore, I suggest major revision of the manuscript based on the following comments:  

1.       In Figure 1c, more discussion about the quality of the graphene sheets should be presented based on the 2D band in Raman spectra. For further help and support see, e.g., [Carbon 81 (2015) 158-166] for rGO and [Nano Lett. 2007, 7, 9, 2645–2649] for pristine graphene.

2.       For the SEM images, please give the same magnifications for the SEM images for better comparison.

3.       The images shown in Figures 3a, 5 and 7 need scale bar.

4.       Figures 4 and 6 should be merged.

5.       Either Figure 5 of Figure 7 should be transferred to supplementary information section. They seem very similar in the results and so one of them should be transferred to the SI.  

6.       The number of biological tests should be given in the caption of the related figures.

7.       The well-known mechanisms involved in the antibacterial activity of nanomaterials are: 1) physical direct interaction of extremely sharp edges of nanomaterials with cell wall membrane [Toxicity of graphene and graphene oxide nanowalls against bacteria], 2) ROS generation [RSC Adv., 2015,5, 80192-80195] even in dark [Langmuir 2015, 31, 33, 9155–9162], 3) trapping the bacteria within the aggregated nanomaterials [Wrapping bacteria by graphene nanosheets for isolation from environment, reactivation by sonication, and inactivation by near-infrared irradiation], 4) oxidative stress [ACS Nano 2011, 5, 9, 6971–6980], 5) interruption in the glycolysis process of the cells [Escherichia coli bacteria reduce graphene oxide to bactericidal graphene in a self-limiting manner], 6) DNA damaging [Free Radical Biology and Medicine Volume 51, Issue 10, 15 November 2011, Pages 1872-1881], 7) heavy metal ion release [ACS Appl. Mater. Interfaces 2014, 6, 4, 2791–2798], and recently 8) contribution in generation/explosion of nanobubbles [Oxygen-Rich Graphene/ZnO2-Ag nanoframeworks with pH-Switchable Catalase/Peroxidase activity as O2 Nanobubble-Self generator for bacterial inactivation]. These mechanisms should be addressed in the revised version and then the dominant mechanism(s) in this work should be highlighted using suitable discussions/supports.

8.       Could the authors comment on the hydrophilicity and also surface charge of the samples? These are important parameters in discussing the antibacterial mechanisms.

9.       The authors should give the optical absorption of the samples especially in the NIR region. GO shows low optical absorption in NIR region. In contrast, rGO can show significant absorption in this region, too. Therefore, the chemical state of the graphene used in the composition should be also evaluated. FTIR and/or XPS can be used in this regard.

10.   It has been mentioned that “Near-infrared irradiation used in photothermal therapy can penetrate deep into the tissue and effectively kill bacteria with little photodamage”. It is right. But, Figures 4 and 6 show that in the best case only ~50% of the bacteria were killed by the photothermal effect. Why? Please clarify this deficiency.

11.   It has been mentioned that “These properties have resulted in a rising number of studies using GO as effective photothermal agents in photothermal therapy”. This needs to be supported by, e.g., [Graphene nanomesh promises extremely efficient in vivo photothermal therapy] & [Biomaterials Volume 33, Issue 7, March 2012, Pages 2206-2214].  

Round 2

Reviewer 1 Report

looks better

Author Response

Thank you.

Reviewer 2 Report

The current version can be accepted.

Author Response

Thank you.

Reviewer 3 Report

The authors tried to revise the manuscript based on the comments. However, there are still significant points which should be clarified, addressed and discussed, as mentioned below:

Concerning Comment #1, the supported discussion on the 2D band is still required.

Concerning Comment #7, the response is “The requested part was added to the review along with the related paper”. But, the revised version shows that the mentioned mechanisms have been not included into the text. This should be considered based on the detailed comment. Moreover, in the third point of this comment there is paper relating to NIR photothermal inactivation of bacteria by graphene. The authors, at first, should clarify the novelty of this work, as compared to the previous published ones.    

The Comment #9 was left nearly without any proper response.

Concerning Comment #11, the response is “The requested part was added to the review along with the related paper”. But, ref. [24] is related to another work of Yang et al. and ref. [25] is related to an in vitro study, rather than an in vivo one. Please consider the comment based on its details. 

Therefore, the manuscript can be considered for further consideration, if the above comments are addressed completely.

Author Response

Response to Reviewer 3(2) Comments

Point1: Concerning Comment #1, the supported discussion on the 2D band is still required.

Response 1: The requested part was added to the paper

It has been shown experimentally [1, 2] that the 2D line in graphene changes in shape, width and position with number of layers. Lucchese et.al [3] and Martins Ferreira et. al [4] have studied the evolution of the Raman spectra for mono and multi-layer graphene with increasing disorder, showing that the intensity of the D line, which is absent in pristine graphene, increases when disorder is induced in the sample up to a maximum value where it begins to decrease. On the other hand, the 2D line intensity is maximum for pristine graphene and it decreases with increasing disorder.[5]

  1. A.C. Ferrari, J.C. Meyer, V. Scardaci, C. Casiraghi, M. Lazzeri, F. Mauri, S. Piscanec, D. Jiang, K.S. Novoselov, S. Roth , and A. K. Geim, Phys. Rev. Let. 97, 187401 (2006).
  2. A. Gupta, G. Chen, P. Joshi, S. Tadigadapa, and P.C. Eklund, Nano Lett. 6, 2667 (2006).
  3. M.M. Lucchese, F. Stavale, E.H. Martins Ferreira, C. Vilani, M.V.O. Moutinho, R.B. Capaz, C.A. Achete and A. Jorio, Carbon 48, 1592 (2010)
  4. E. H. Martins Ferreira, M. V. O. Moutinho, F. Stavale, M. M. Lucchese, R. B. Capaz, C. A. Achete and A. Jorio, Phys. Rev. B 82, 125429 (2010).
5. Pedro Venezuela, Michele Lazzeri, and Francesco Mauri,Phys. Rev. B 84, 035433 – Published 25 July 2011

Point2: Concerning Comment #7, the response is “The requested part was added to the review along with the related paper”. But, the revised version shows that the mentioned mechanisms have been not included into the text. This should be considered based on the detailed comment. Moreover, in the third point of this comment there is paper relating to NIR photothermal inactivation of bacteria by graphene. The authors, at first, should clarify the novelty of this work, as compared to the previous published ones.    

Response 2: The requested part was added to the paper

There are several mechanisms for bacterial death of graphene oxide. The antibacterial mechanism of grahene known so far is the physical destruction of the cell membrane and oxidative stress damage. [46,47] It is known that reactive oxygen species (ROS)-mediated oxidative stress is generated by graphene, which causes serious damage to bacterial cells and has antibacterial effect. [48,49] Another antibacterial process is the dispersibility and trapping ability of oxygen-containing functional groups of GO. [47]. Jang et al. (2021) reported that attachment of S.mutans was reduced by 58.58% and the biofilm thickness by 43.49% in the group Zr-GO. [50] And graphene also showed excellent photothermal performance in response to NIR light. It is found that a small amount of ROS could enhanced the thermal susceptibility of bacteria to decrease the required temperature for elimination, while hyperthermia can disrupt the bacterial membrane to facilitate the permeation of ROS into bacteria. [1] Photothermal bactericidal surfaces based on hyperthermia provide a biocide-free and high-efficiency way to solve the problems related to surface-attached bacteria, in particular, the multidrug-resistant bacteria. The bacteria-killing mechanism of this thermal effect was resulted from both the hyperthermia-induced protein denaturation and subsequent bacterial thermal decomposition. [2] Although the remarkable achievements of photothermal bactericidal surfaces have been made, it should be noted that there are still some challenges related to this promising field and will be left. In this study, graphene was coated using plasma to overcome these disadvantages. Plasma is a gas that is ionized and charged with energy and treatment of living tissues with plasma can change the wettability, as well as their mechanical and biological properties. [3] The proposed method of coating graphene with plasma has the advantage of being simple and reasonable in terms of cost, while not requiring other additives or generating by-products during the production of GO. [4] With this surface coating method, graphene oxide without deformation was deposited on the surface simply and quickly. In the results, the adhesion of S. mutans and P. gingivalis was not significant reduced when comparing the Z and ZG groups. However, when the ZG group and the ZGN group were compared, a significant reduction was obtained.

  1. M. Yin, Z. Li, E. Ju, Z. Wang, K. Dong, J. Ren and X. Qu, Chem. Commun., 2014, 50, 10488- 10490.
  2. Y. Zou, Y. zhang, Q. Yu and H. Chen, Biomater. Sci., 2020, DOI: 10.1039/D0BM00617C
  3. Chu P, Plasma-surface modification of biomaterials. Materials Science and Engineering: R: Reports, 2002;36(5-6):143-206.
  1. Rho KH, Park C, Alam K, Kim DY, Ji MK, Lim HP, Ch HS, Biological effects of Plasma-based graphene oxide deposition on Titanium. Journal of Nanomaterials, 2019;(3);1-7.

Point3: The Comment #9 was left nearly without any proper response

Response 3: The thesis was written with graphene oxide, but I will clarify that the experiment was conducted by coating with rGO, which is specifically hydrophobic. As a result of the surface treatment, it was confirmed that it was more hydrophobic than the control group. The paper will be discussed with a clear distinction between rGO and GO, and rGO with higher light-heat absorption will be additionally written in the discussion.

The requested part was added to the paper

Reduced graphene oxide(rGO) is a promising alternative for bulk production of graphene-like materials. The bottleneck of its commercialization is the control of oxygen functional groups on the surface to engineer its diverse properties, such as electronic structure, optical properties, and surface properties. [*]Both Graphene oxide(GO) and reduced graphene oxide (rGO) effectively absorb near-infrared (NIR) light, which is a biocompatible light source that penetrates tissues. Moreover, GO and rGO convert the absorbed NIR light energy to heat, increasing the temperature in GO and rGO and their surrounding media. [1,2] While both GO and rGO can absorb NIR, rGO is more effective [3] likely because of the red shift in the absorbance peak from approximately 230 to 260 nm [1, 4]. In related study, RGO synthesized in this study showed about 10 times higher absorbance than GO at 880 nm. The reason for the small temperature rise of GO is that the absorbance at 800 nm is significantly lower than that of other samples. [5] In this paper, after coating rGO on the surface using plasma, oral bacteria were tested.

* Pei, S.; Cheng, H.-M. The reduction of graphene oxide. Carbon 201250, 3210–3228.

1.Hashemi M, Omidi M, Muralidharan B, Smyth H, Mohagheghi MA, Mohammadi J, et al. Evaluation of the photothermal properties of a reduced graphene oxide/arginine nanostructure for near-infrared absorption. ACS Appl Mater Interfaces. 2017;9(38):32607–20.

  1. Shang J, Ma L, Li J, Ai W, Yu T, Gurzadyan GG. The origin of fluorescence from graphene oxide. Sci Rep. 2012;2(1):1–8.
  2. Yang K, Wan J, Zhang S, Tian B, Zhang Y, Liu Z. The influence of surface chemistry and size of nanoscale graphene oxide on photothermal therapy of cancer using ultra-low laser power. Biomaterials. 2012;33(7):2206–14.

4.Shi H, Wang C, Sun Z, Zhou Y, Jin K, Redfern SA, et al. Tuning the nonlinear optical absorption of reduced graphene oxide by chemical reduction. Opt Express. 2014;22(16):19375–85.

5.Seunghwa Lee* and So Yeon Kim, Preparation and Characterization of Reduced Graphene Oxide with Carboxyl Groups-Gold Nanorod Nanocomposite with Improved Photothermal Effec, Appl. Chem. Eng., Vol. 32, No. 3, June 2021, 312-319 https://doi.org/10.14478/ace.2021.1034

Point4: Concerning Comment #11, the response is “The requested part was added to the review along with the related paper”. But, ref. [24] is related to another work of Yang et al. and ref. [25] is related to an in vitro study, rather than an in vivo one. Please consider the comment based on its details

24,25 빼기

Response 4: The requested part was added to the paper

Gulzar et al. [1] used both photodynamic and photothermal therapies against cancer cells by conjugating Chlorin e6 to up conversion nanoparticles that were then conjugated to GO. Singlet oxygen was generated alongside an increase in temperature under 808 nm irradiation which was successfully used in in vivo tumor treatment.

1 Gulzar A, Xu J, Yang D, Xu L, He F, Gai S, et al. Nano-graphene oxide-UCNP-Ce6 covalently constructed nanocomposites for NIR mediated bioimaging and PTT/PDT combinatorial therapy. Dalton, Trans. 2018;47(11):3931–9.

Round 3

Reviewer 3 Report

The authors prepared some responses to the comments. But, it seems that the authors are using another version of the first comments. Therefore, I refer the authors to careful consideration of the first version of the comments. The main points which should be clarified are as follows:

Concerning Comment #1, the position, shape and relative intensity of the 2D band should be used in analyzing the quality of the graphene contents, using suitable supports previously mentioned.

Concerning comment #7, the possible well-known mechanisms should be addressed accordingly and discussed.

Comment #9 is still leaved without a suitable response. This comment is related to optical absorption of GO especially in the NIR region rather the hydrophilicity.

Concerning Comment #11, the response seems vague and irrelevant.

The manuscript cannot be suggested for publication unless these items will be clarified points by points. At this stage, I can only suggest major revision of the manuscript (boarder line to rejection) by fully addressing to the first version comments.